# Decoupled Marked Temporal Point Process using Neural ODEs

**Yujee Song** [†]     **Lee Dong Hyun**[†]     **Rui Meng**[‡]     **Won Hwa Kim**[†]

[†] Pohang University of Science and Technology     [‡]Amazon.com Inc.

{yujees,dongtak,wonhwa}@postech.ac.kr    rmmeng@amazon.com

## Abstract

A Marked Temporal Point Process (MTPP) is a stochastic process whose realization is a set of event-time data. MTPP is often used to understand complex dynamics of asynchronous temporal events such as money transaction, social media, healthcare, etc. Recent studies have utilized deep neural networks to capture complex temporal dependencies of events and generate embeddings that aptly represent the observed events. While most previous studies focus on the inter-event dependencies and their representations, how individual events influence the overall dynamics over time has been under-explored. In this regime, we propose a Decoupled MTPP framework that disentangles characterization of a stochastic process into a set of evolving influences from different events. Our approach employs Neural Ordinary Differential Equations (Neural ODEs) (Chen et al., 2018) to learn flexible continuous dynamics of these influences while simultaneously addressing multiple inference problems, such as density estimation and survival rate computation. We emphasize the significance of disentangling the influences by comparing our framework with state-of-the-art methods on real-life datasets, and provide analysis on the model behavior for potential applications.

## 1 Introduction

Event-time data, which are found across various practical domains such as social media (Zhao et al., 2015; Leskovec & Krevl, 2014), healthcare (Du et al., 2016; Meng et al., 2023) and stock changes (Bacry et al., 2015; Hawkes, 2018), comprise timelines of events of diverse types. Modeling such temporal data as a stochastic process, one seeks to predict time and type of the future events based on the history, i.e., previously observed sequential events. For example, a history of purchases of a person may tell when the person will buy a new item, and a short message from a famous social media influencer could affect a critical bull or bear in a stock market. Predicting such a future event is often realized as a chance of the event, i.e., a likelihood of the event $e$ at a specific time $t$, which constitutes a probability distribution function (pdf) $f(e, t)$.

Classically, the event prediction has been handled using a Temporal Point Process (TPP), which is a stochastic process whose realization is a set of ordered time (Rasmussen, 2018). TPP characterizes at which time point an event will occur given historical time points. Hawkes Process effectively models this stochastic behavior by defining an *conditional intensity function* which leverages the sum of excitation functions induced by past events called "self-excite" characteristics (Hawkes, 1971; Adamopoulos, 1976). The Hawkes Process independently models the influence of each event and incorporates the entire event cases using a simple linear operation (i.e., summation) of the influences offering interpretability.

While various models for TPP are adept at modeling dynamically forthcoming events, they do not incorporate event-related details such as location, node-specific information (particularly when observations pertain to graphs such as multi-sensor or social networks), categorical data, and contextual information (e.g., tokens, images, or textual descriptions). Marked Temporal Point Process (MTPP) addresses it with "marks" which denotes the type of events, which elevates the traditional TPP of a univariate event to multiple marks (i.e., types) of the event. Many recent approaches for the MTPP leverage combinations of the intensity function in the Hawkes Process with neural networks to flexibly characterize the probability density function (pdf) of marks (Mei & Eisner, 2017; Jia & Benson, 2019; Zhang et al., 2020; Chen et al., 2021; Yang et al., 2022). This line of works includes Recurrent

Marked Temporal Point Process (Du et al., 2016), Neural Hawkes Process (Mei & Eisner, 2017) and Transformer Hawkes Process (Zuo et al., 2020) that effectively models the behavior of MTPP.

The aforementioned approaches successfully handled the complicated dependencies of intensity functions with the neural network, however, they come at the cost of interpretability of the dependencies among events. Existing methods often directly estimate the probability of the event across time and marks as a whole, or the intensity function as a proxy is approximated without considering the dynamics of influences from various marks and time. Moreover, reconstruction of the pdf of marks from the intensity function as well as other characteristic functions such as survival function require several integration operations without a closed-form solution, which must be re-evaluated whenever a new event is considered during inference resulting in exponential increase in computation time.

In this regime, we introduce a novel decoupled MTPP framework characterized by decoupled hidden state dynamics, utilizing latent continuous dynamics driven by neural ordinary differential equations (ODE), named as Dec-ODE. The hidden state dynamic is modeled separately across each event, and it explains how each event separately contributes to the overall stochastic process with high fidelity. Moreover, such a decoupling approach equipped with the ODE lets our model train on the continuous dynamics of the hidden states in parallel, which results in substantial decrease in computation time compared to previous approaches that sequentially compute the dynamics from a series of events. The formulation of Dec-ODE leads to the contributions summarized as follows:

- We propose a novel MTPP modeling framework, i.e., Dec-ODE, which decouples individual events within a stochastic process for efficient learning and explainability of each mark,

- Dec-ODE models the continuous dynamics of influences from individually decoupled events by leveraging Neural ODEs,

- Dec-ODE demonstrates state-of-the-art performance on various benchmarks for MTPP validation while effectively capturing inter-time dynamics with interpretability.

## 2 BACKGROUND

**Temporal Point Process (TPP)** is a stochastic process whose realization is a set of ordered times $\{t_i\}_{i=0}^N$. The observed events until time $t_i$ is denoted as a history $\mathcal{H}_{t_i}$. The time of the next event, given $\mathcal{H}_{t_i}$, can be characterized using a pdf $f(t|\mathcal{H}_{t_i}) := f^*(t)$, which denotes a probability that a subsequent event occurs during the interval $[t, t + dt)$ conditioned on $\mathcal{H}_{t_i}$. Its cumulative density function (cdf) is denoted as $F(t|\mathcal{H}_{t_i})$, and a survival function is denoted as $S(t|\mathcal{H}_{t_i}) = 1 - F(t|\mathcal{H}_{t_i})$, where both are conditioned on $\mathcal{H}_{t_i}$. Throughout this paper, $*$ will be used to state that a function is conditioned on $\mathcal{H}_{t_i}$.

It is often difficult to define a fixed functional form for $f^*(t)$ due to its unknown pattern and the constraint $\int_0^\infty f^*(s)ds = 1$. Therefore, an intensity function $\lambda^*(t) := \frac{f^*(t)}{S^*(t)} = \frac{f^*(t)}{1 - F^*(t)}$ is often introduced for the characterization of the stochastic process (Rasmussen, 2018). The $\lambda^*(t) \geq 0$ being the only constraint, it has been commonly used to model a TPP with flexibility(Hawkes, 1971; Mei & Eisner, 2017; Omi et al., 2019; Jia & Benson, 2019; Zhang et al., 2020; Zuo et al., 2020; Chen et al., 2021; Yang et al., 2022). A conditional intensity function, $\lambda^*(t)$, utilizes observed events $\mathcal{H}_t$ to calculate the intensity at $t$, and $\lambda^*(t)dt = p(t_i \in [t, t + dt)|\mathcal{H}_t)$.

**Marked Temporal Point Process (MTPP)** is a random process whose realization is a set of events $\mathcal{S} = \{e_i\}_{i=0}^N$, where $e_i = (t_i, k_i)$. Here, the $k_i \in \{1, \ldots, K\}$ denotes a class of the event, i.e., a mark, and $t_i \in \mathbb{R}$ is the time of its occurrence with $t_i < t_{i+1}$. The entire sequence prior to time $t_i$ is denoted as the history $\mathcal{H}_{t_i} = \{(t_j, k_j) \in \mathcal{S} : t_j < t_i\}$. The conditional intensity function $\lambda^*(t, k) := \lambda(t, k|\mathcal{H}_t)$ for MTPP with a mark $k$ is often conveniently written as a product of ground intensity $\lambda_g^*(t)$ and conditional intensity of marker $f^*(k|t)$ given time as (Rasmussen, 2018; De et al., 2019; Daley & Vere-Jones, 2003):

$$\lambda^*(t, k) = \lambda_g^*(t)f^*(k|t)$$
$$= \frac{f(t|\mathcal{H}_t)f^*(k|t)}{1 - F(t|\mathcal{H}_t)} = \frac{f(t, k|\mathcal{H}_t)}{1 - F(t|\mathcal{H}_t)} \tag{1}$$

where $f^*(t)$ is the probability with respect to time, and $f^*(k|t)$ is the probability with respect to mark. Since $\lambda^*(t, k)$ can be expressed in two separate components $\lambda_g^*(t)$ and $f^*(k|t)$, the likelihood of the MTPP can be expressed as follows (Daley & Vere-Jones, 2003):

$$f(t, k) = \left[ \prod_{i=1}^{\mathcal{N}_g(t_N)} \lambda_g^*(t_i) \right] \left[ \prod_{i=1}^{\mathcal{N}_g(t_N)} f^*(k_i|t_i) \right] \exp\left( -\int_0^{t_N} \lambda_g^*(u)\, du \right). \tag{2}$$

Here, the $\mathcal{N}_g$ is a counting process characterized by the *ground intensity function* $\lambda_g^*(t)$, where the mark of the events is ignored.

**Hawkes Process** is widely utilized for modeling point processes with self-excitatory behavior. It describes a TPP, where each event triggers subsequent events. Formally, the Hawkes process is defined by an intensity function $\lambda^*(t)$ as (Adamopoulos, 1976):

$$\lambda^*(t) = v + \sum_{t_i < t} \mu(t - t_i) \tag{3}$$

where $t$ denotes the time of interest, $v$ is a non-negative constant, $\mu(t)$ is an *excitatory function* that models the impact of past events on future event rates, and $t_i$ denotes past event time points. The Hawkes process has been broadly studied in machine learning (Yang et al., 2022; Mei & Eisner, 2017; Zuo et al., 2020; Zhang et al., 2020) with practical applications (Hawkes, 2018; Cai et al., 2018; Du et al., 2015).

**The Neural Ordinary Differential Equations** (Neural ODEs) blends neural networks with differential equations, offering an interesting approach to deep learning (Chen et al., 2018) by employing continuous-depth models instead of fixed-layer architecture. Mathematically, they define continuous transformations of network outputs governed by ordinary differential equations (ODEs):

$$\frac{d\mathbf{z}(t)}{dt} = \gamma(\mathbf{z}(t), t|\theta) \tag{4}$$

where $\gamma(\cdot|\theta)$ is a neural network parameterized by $\theta$ to approximate changes of a hidden state $\mathbf{z}(t)$ with respect to time $t$. In our framework, the neural ODEs is employed to model the continuous dynamics of how each event affects the overall MTPP.

## 3 DECOUPLING MARKED TIME POINT PROCESS

Our approach focuses on establishing a general framework that captures the high-fidelity distribution of MTPP with decoupled structures. Specifically, given a sequence of events denoted as $\mathcal{S} = \{e_i\}_{i=0}^n$, where an event $e_i = (t_i, k_i)$ is composed of a time point and a marker, with $n + 1$ observed events, our goal is to predict a probability distribution of an event $e_{n+1}$. Unlike previous works that directly estimate the conditional intensity $\lambda^*(t, k)$, we take an alternative approach of separately estimating the ground intensity and the distribution of event, i.e., $\lambda_g^*(t)$ and $f^*(k|t)$. In our framework, the influences from each events through time are represented by hidden state dynamics $h(t; e_i)$, which undergo a decoding process to yield $\lambda_g^*(t)$ and $f^*(k|t)$ separately. As described in Fig.1, $\lambda_g^*(t)$ and $f^*(k|t)$ form independent trajectories, and they are combined in the later stage to constitute $\lambda^*(t, k)$.

### 3.1 HIDDEN STATE DYNAMICS

In this section, we introduce decoupled hidden state dynamics, where the hidden state is used to characterize the influence each event $e_i$ has on the overall process. The decoupled hidden states $h(t; e_i) \in \mathbb{R}^d$, where $i = 1, \cdots, n$, are independently propagated through time from the occurrence of each event at $t_i$. The dynamics of $h(t; e_i)$, which will be used to characterize the complex dynamics of the intensity function within MTPP, are trained using the Neural ODEs (Chen et al., 2018).

Specifically, the initial magnitude of a hidden state $h(t_i; e_i)$ depends on the event type $k_i$, and the dynamics are solved via initial value problem (IVP) solvers. Then the hidden state caused by an

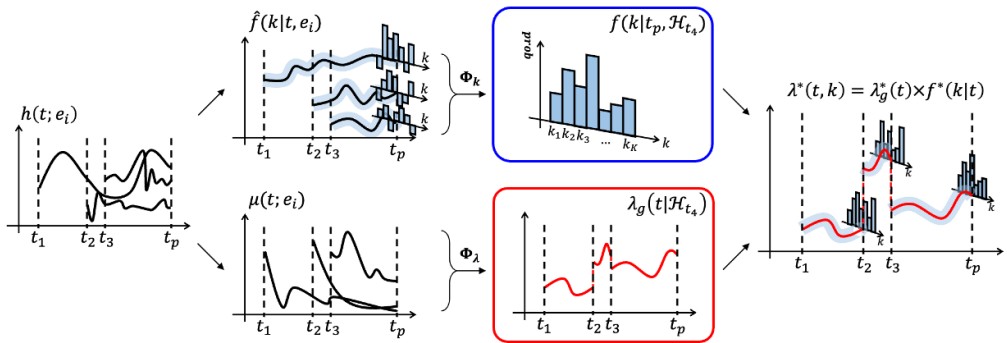

Figure 1: A visualization of the overall framework. Hidden states from each event $e_i$ are independently propagated and decoded into trajectories of $\mu(t; e_i)$ and $\hat{f}_k(k|t, e_i)$. Each trajectory represents the effect of each event on the MTPP, and the MTPP can be reconstructed by combining all trajectories.

event $e_i$ at time $t$ can be obtained as:

$$dh(t; e_i) = \gamma(h(t; e_i), t, k_i; \theta)dt \tag{5}$$

$$h(t; e_i) = h(t_i; e_i) + \int_{t_i}^{t} \gamma(h(s; e_i), s, k_i; \theta)ds$$

$$= W_e(k_i) + \int_{t_i}^{t} \gamma(h(s; e_i), s, k_i; \theta)ds \tag{6}$$

where the initial magnitude is obtained using $W_e(\cdot) : \{1, \cdots, K\} \rightarrow \mathbb{R}^d$, which is trainable, and $\gamma$ is parameterized with a neural network $\theta$. Hidden states at time $t$ can be computed in parallel as a multidimensional ODE by taking advantages of the decoupled structure of $h(t; e_i)$ as

$$\frac{d}{dt}\mathbf{h(t)} = \frac{d}{dt} \begin{bmatrix} h(t; e_0) \\ \vdots \\ h(t; e_i) \end{bmatrix} = \begin{bmatrix} \gamma(h(t; e_0), t, k_0; \theta) \\ \vdots \\ \gamma(h(t; e_i), t, k_i; \theta) \end{bmatrix}. \tag{7}$$

One major advantage of the decoupling is that the influence of events from $\mathbf{h}(t)$ can be selectively considered. For example, when computing $f^*(t)$ conditioned on $\mathcal{H}_{t_j}$, where $t > t_{j+1}$, $h(t; e_{t_{j+1}})$ should not be included in computation. The details will be discussed in the Sec. 3.2.

## 3.2 GROUND INTENSITY FUNCTION

In our method, the ground intensity $\lambda_g^*(t)$ is defined as a mixture of decoupled *influence functions* $\mu(t; e_i)$, which can be considered as a generalized form of the exciting functions found in the Hawkes process (Adamopoulos, 1976). Since each influence function is conditioned on a single event, in order to define $\lambda_g(t|\mathcal{H}_t)$, i.e., the ground intensity conditioned on the history, the influences from historical events must be aggregated. The conditional ground intensity function is defined as,

$$\lambda_g(t|\mathcal{H}_{t_{n+1}}) = \lambda_g(t|e_0, \cdots, e_n) = \Phi_\lambda(\mu(t; e_0), \cdots, \mu(t; e_n)) \tag{8}$$

where $\mu(t; e_i) := g_\mu(h(t; e_i))$ is decoded by a neural network $g_\mu : \mathbb{R}^d \rightarrow \mathbb{R}$, $e_i \in \mathcal{H}_{t_{n+1}}$ is an observed event, and $\Phi_\lambda$ is a positive function to satisfy the non-negativity constraints of $\lambda_g^*(t)$. Notice that the hidden state $h(t; e_i)$ is decoded into the influence function $\mu(t; e_i)$ before aggregated into $\lambda_g^*(t)$, otherwise, influence from a specific event would not be observable.

When learning a TPP, integration is pivotal. Not only does the computation of probability distribution $f^*(t) := \lambda^*(t) \exp(-\int_{t_{i-1}}^{t} \lambda^*(s)ds)$ requires an integration, but also obtaining characteristic functions, survival rate, etc. requires additional integration which does not have a closed-form solution in general. Therefore, intensity-based methods often require additional steps for predictions. Methods such as Monte Carlo Inference (Mei & Eisner, 2017; Zhang et al., 2020; Zuo et al., 2020; Yang et al., 2022) are often used for calculating $f^*(t)$, while sampling methods such as the thinning algorithm Mei & Eisner (2017) are adopted for computing the expected value. In these approaches, each integration has to be handled separately with additional sampling. The Neural ODEs, which we

extensively utilize to solve for the MTPP, is inherently computationally heavier compared to Monte Carlo Inference due to its sequential solving mechanism. To circumvent the extra computational burden, we leverage the characteristics of the differential equation. By simultaneously solving the following multi-dimensional differential equation with respect to $\mathbf{h}(t)$ along with numerical integration, we eliminate the need for additional sampling:

$$\frac{\partial}{\partial t} \begin{bmatrix} \mathbf{h}(t) \\ \Lambda_g(t|\mathcal{H}_{t_i}) \\ F(t|\mathcal{H}_{t_i}) \\ \mathbb{E}[t] \end{bmatrix} = \begin{bmatrix} \gamma(\mathbf{h}(t), t, \mathbf{k}; \theta) \\ \lambda_g(t|\mathcal{H}_{t_i}) \\ f(t|\mathcal{H}_{t_i}) \\ t \cdot f(t|\mathcal{H}_{t_i}) \end{bmatrix} = \begin{bmatrix} \gamma(\mathbf{h}(t), t, \mathbf{k}; \theta) \\ \Phi_\lambda(g_\mu(\mathbf{h}(t))) \\ \lambda_g(t|\mathcal{H}_{t_i}) \cdot \exp(\Lambda_g(t_{i-1}|\mathcal{H}_{t_i}) - \Lambda_g(t|\mathcal{H}_{t_i})) \\ t \cdot f(t|\mathcal{H}_{t_i}) \end{bmatrix} \quad (9)$$

where $\Lambda_g(t|\mathcal{H}_{t_i}) := \int_0^t \lambda_g(t|\mathcal{H}_{t_i})dt$ is called the *compensator*, and $\mathbf{k}$ is the marks corresponding to $\mathbf{h}(t)$. Because the dynamics are learned through differential equations, values at time $t$ can be used for computing others. e.g., $\lambda_g^*(t)$, which is a derivative of $\Lambda_g^*(t)$, can be computed using $\mathbf{h}(t)$.

Therefore, important estimations with integration, such as likelihood, survivor rate $S^*(t) := 1 - F^*(t)$, and $\mathbb{E}_{f^*}[t]$ can be predicted in a single run. This formulation can be applied to any number of integrations. Moreover, as briefly mentioned in Sec. 3.1, the influence functions can be selected according to our interests as illustrated in Fig. 2. Therefore, approximations of functions in equation 9 under different $\mathcal{H}_{t_i}$ can be obtained in parallel, distinguishing Dec-ODE from other intensity-based methods that require separate runs.

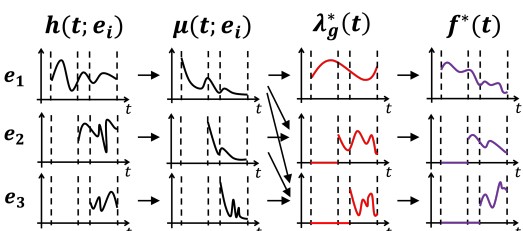

Figure 2: Visualization of equation 9. Different combinations of $\mu(t; e_i)$ can be selected for calculating $\lambda_g^*(t)$ and $f^*(t)$ conditioned on different $\mathcal{H}_{t_i}$ in parallel.

### 3.3 CONDITIONAL PROBABILITY FOR MARKS

Similar to the formulation of the ground intensity in equation 8, the probability of marks $f^*(k|t)$ can be expressed using the following form:

$$f(k|t, \mathcal{H}_{t_{n+1}}) = \Phi_k(\hat{f}(k|t, e_0), \cdots, \hat{f}(k|t, e_n)) \quad (10)$$

where $\hat{f}(k|t, e_i) := g_f(h_t(e_i))$ is an influence from $e_i$ on $f(k|t)$, decoded by a neural network $g_f(\cdot)$. The function $\Phi_k$ combines $\hat{f}(k|t, e_i)$ over events while satisfying $\sum_K \Phi_k(\cdot) = 1$.

The intuition behind such a modeling is that the context of an event changing over time carries important information. For instance, getting a driver's license would dramatically increase the probability of causing a car accident since there was no chance without the license. However, as time goes on, the person gets better at driving and the probability decreases. Using the proposed structure, the changes of implications through time can be inferred by $\hat{f}(k|t_i, e_i)$.

## 4 LINEAR DEC-ODE

The combining functions $\Phi_\lambda$ and $\Phi_k$ can be chosen from a simple summation to a complex neural networks, such as Transformer. Nonetheless, in the following section, we present a linear version of Dec-ODE as it offers efficiency and interpretability via parallel modeling of the hidden states.

### 4.1 COMBINING INFLUENCES FROM PAST EVENTS

In order to demonstrate the competence of the proposed framework, the implementation of $\Phi_\lambda$ and $\Phi_k$ is defined as linear and simple equations that combines influences of historical events for $\lambda_g^*(t)$ and $f^*(k|t)$. The ground intensity $\lambda_g^*(t)$ and the conditioned probability of marks $f^*(k|t)$ are defined as combinations of decoupled dynamics :

$$\lambda_g(t|\mathcal{H}_{t_{n+1}}) = \Phi_\lambda(\mu(t; e_0), \cdots, \mu(t; e_n)) = \sum_{e_i \in \mathcal{H}_{t_{n+1}}} \text{softplus}(\mu(t; e_i)) \quad (11)$$

$$f(k|t, \mathcal{H}_{t_{n+1}}) = \Phi_k(\hat{f}(k|t, e_0), \cdots, \hat{f}(k|t, e_n)) = \text{softmax}\left( \sum_{e_i \in \mathcal{H}_{t_{n+1}}} \hat{f}(k|t, e_i) \right) \quad (12)$$

where softplus and softmax functions are used to satisfy the constraints from Sec. 3.2 and Sec. 3.3, respectively. This modeling of $\lambda_g^*(t)$ with linear summation resembles with the Hawkes process. Yet, our method can model more flexible temporal dynamics, and better suited for modeling complex real-life scenarios.

## 4.2 TRAINING OBJECTIVE

The objective of our method is to maximize the likelihood of the predicted Marked TPP (Daley & Vere-Jones, 2003). Taking a $\log$ on the likelihood in equation 2,

$$
\begin{aligned}
\ln f(t, k) &= \ln \left[ \prod_{i=1}^{\mathcal{N}_g(t_N)} \lambda_g^*(t_i) \right] \left[ \prod_{i=1}^{\mathcal{N}_g(t_N)} f^*(k_i|t_i) \right] \exp\left( -\int_0^{t_N} \lambda_g^*(u) du \right) \\
&= \underbrace{\sum_{i=1}^{\mathcal{N}_g(t_N)} \ln \lambda_g^*(t_i) - \int_0^{t_N} \lambda_g^*(u) du}_{\ln L_\lambda} + \underbrace{\sum_{i=1}^{\mathcal{N}_g(t_N)} \ln f^*(k_i|t_i)}_{\ln L_k}
\end{aligned}
\quad (13)
$$

where $t_N$ is the last observed time point, and the log-likelihood for the ground intensity $\ln L_\lambda$ and the mark distribution $\ln L_k$ are independently defined, and used to learn $\lambda_g(t)$ and $f(k|t)$, respectively. Intuitively, the first term in $L_\lambda$ is the probability of event happening at each $t_i$ and the second term is the probability of event not happening everywhere else. Therefore, $\lambda_g^*(t)$ should be high at the time of event's occurrence and low everywhere else to maximize $L_\lambda$. Also, for the estimations, $f(t)$ is normalized to satisfy $\int f(t)dt = 1$.

## 4.3 TRAINING SCHEME

Many previous works using differential equation based modeling (Jia & Benson, 2019; Chen et al., 2021) sequentially solve the entire time range, which require exhaustive training time. In our case, using the characteristics of $\Phi_\lambda$ with linear summation, such limitation can be alleviated as follows.

First, because each $\mu(t; e_i)$ is independent from other events, we can propagate them simultaneously as a multi-dimensional differential equation as

$$d \begin{bmatrix} \mu(\tau_0; e_0) \\ \vdots \\ \mu(\tau_i; e_i) \end{bmatrix} = \begin{bmatrix} \gamma(\mu(\tau_0), \tau_0, k_0; \theta) \\ \vdots \\ \gamma(\mu(\tau_i), \tau_i, k_i; \theta) \end{bmatrix} \cdot d\boldsymbol{\tau} \quad (14)$$

where $\boldsymbol{\tau} = [\tau_0, \cdots, \tau_i]^\top = [t_0 + t, t_1 + t, \cdots, t_i + t]^\top$.

Second, one of the benefits of formulating the ground intensity as a linear equation is that the compensator $\Lambda_g^*(t)$ can also be calculated in the same manner as in equation 11

$$\Lambda_g^*(t) = \int_0^t \lambda_g^*(s)ds = \int_0^t \sum_{e_i \in \mathcal{H}_t} \text{softplus}(\mu(s; e_i))ds \quad (15)$$

$$= \sum_{e_i \in \mathcal{H}_t} \int_{t_i}^t \text{softplus}(\mu(s; e_i))ds \quad (16)$$

where the region of the integration changes since there is no influence of $e_i$ before $t_i$. The equation 16 conveys that the integration in equation 13 can be computed by integrating $\text{softplus}(\mu(t; e_i))$ individually first and sum up later, instead of sequentially going through the entire sequence. Hence, if we can find the number of time-points $m$, we can solve the entire trajectory within a fixed number of steps. The effectiveness of such modeling is discussed in Sec. 6.4.

## 5 RELATED WORKS

**Intensity modeling.** When modeling TPP, tdhe intensity function is widely used due to its flexibility in capturing temporal distributions, and deep learning facilitated the parameterization of complex

processes using neural networks. Initially, many focused on relaxing constraints of traditional TPP (Hawkes, 1971; Isham & Westcott, 1979; Daley & Vere-Jones, 2003) such as Hawkes process. RNN based neural TPP models (Mei & Eisner, 2017; Du et al., 2016) used RNNs to effectively encapsulate sequences into a history vector. Transformer-based methods (Zuo et al., 2020; Zhang et al., 2020) were used to encapsulate $\mathcal{H}_t$ while considering long term and short term dependency of events, and Attentive Neural Hawkes Process (ANHP) (Yang et al., 2022) utilized self-attention for modeling complex continuous temporal dynamics by using a attention network on an unobserved event. Pan et al. (2021) used the Gaussian process to relax the exponential decaying effect of Hawkes process. Omi et al. (2019) adopted a compensator function, which is a integration of an intensity function, to model TPP to alleviate computing issues in integration with no closed form.

**Direct estimation of pdf.** An alternative approach, which is to directly predicting the pdf of a TPP, has also been studied. Shchur et al. (2020) proposed a method to model the temporal distribution using a log-normal mixture. Having a closed form pdf allowed direct inference of important characteristics, while the mixture model provided flexible modeling. There exists various other methods that model the pdf directly using popular deep learning models including GAN, normalizing flow, variational auto encoder and meta learning (Lin et al., 2022; Bae et al., 2023).

**Learning temporal dynamics using Neural ODEs.** In (Chen et al., 2018), Neural ODE was proposed where the changes of hidden vector between layers are considered in a continuous manner. Differential equation based methods were incorporated into various temporal dynamic modeling (Jin et al., 2022; Seedat et al., 2022; Park et al., 2023), showing advantages in modeling sparse and irregular time series. Jia & Benson (2019) modeled the jump stochastic differential equation using Neural ODEs, and tested on TPP modeling. Chen et al. (2021) expanded the work to modeling spatio-temporal point process and predicted both temporal and spatial dynamics using Neural ODEs and continuous normalizing flow. Also, an algorithm for handling time varying sequence in batch wise computation was proposed.

## 6  EXPERIMENT

To evaluate Dec-ODE, we performed experiments under various settings to understand the model behavior. In Sec. 6.2, we compare linear Dec-ODE with the state-of-the-art methods using conventional benchmarks. The explainability of Dec-ODE is discussed in Sec. 6.3, and the parallel computation scheme introduced in Sec. 4.3 is tested in Sec. 6.4.

### 6.1  EXPERIMENT SETUP

**Datasets.** We used five popular real-world datasets from various domains: **Reddit** is a sequences of social media posts, **Stack Overflow (SO)** is a sequences of rewards received from the question-answering platform, **MIMIC-II** is a sequences of diagnosis from Intensive Care Unit (ICU) clinical visits, **MOOC** is a sequence of user interactions with the online course, and **Retweet** is also a sequences of social media posts. See Appendix for detailed description of these datasets. To obtain the results in a comparable scale, all time points were scaled down with the standard deviation of time gaps, $\{t_i - t_{i-1}\}_{i=1}^n$, of the training set similar to Bae et al. (2023).

**Metrics.** We used three conventional metrics for the evaluation: 1) *negative log-likelihood* (NLL) validates the feasibility of the predicted probability density which is calculated via equation 13, 2) *root mean squared error* (RMSE) measures the reliability of the predicted time points of events, 3) *accuracy* (ACC) measures the correctness of the mark distribution $f^*(k|t)$. For the prediction, the expected value $\mathbb{E}_{f^*}[t]$ was used as the next arrival time of an event, and the mark with the highest probability at time $t$, i.e., $\arg\max(f^*(k|t))$, was used as the predicted mark.

**Baseline.** We compared our methods with three state-of-the-art methods: Transformer Hawkes Process (THP) (Zuo et al., 2020), Intensity-Free Learning (IFL) (Shchur et al., 2020), and Attentive Neural Hawkes Process (ANHP) (Yang et al., 2022). As THP and ANHP are both transformer-based methods, they effectively leverage the entire observed events $\mathcal{H}_{t_n}$ for predicting $e_{n+1}$. IFL directly predicts $f^*(t, k) = f^*(t)f^*(k|t)$ using a log-normal mixture to predict a flexible distribution. Since it parameterizes the distribution $f^*(t)$, $\mathbb{E}_{f^*}[t]$ can be obtained in a closed form. More details about the implementation are in Sec.B.3.

Table 1: Comparison with the state of the art methods on 5 popular real-life datasets. Results with **boldface** and underline represent the best and the second-best results, respectively. Following (Zuo et al., 2020; Yang et al., 2022) the mean and std are gained by bootstrapping 1000 times.

| | RMSE | | | | | ACC | | | | | NLL | | | | |
|---|---|---|---|---|---|---|---|---|---|---|---|---|---|---|---|
| | RMTPP | THP | IFL | ANHP | Dec-ODE | RMTPP | THP | IFL | ANHP | Dec-ODE | RMTPP | THP | IFL | ANHP | Dec-ODE |
| MOOC | 0.473 (0.012) | 0.476 (0.010) | 0.501 (0.012) | 0.470 (0.019) | **0.467** (**0.012**) | 20.98 (0.29) | 24.49 (0.22) | 32.30 (1.30) | 31.53 (0.20) | **42.08** (**0.44**) | −0.315 (0.031) | 0.733 (0.047) | **-2.895** (**0.031**) | −2.632 (0.043) | −2.289 (0.191) |
| Reddit | 0.953 (0.016) | 6.151 (0.195) | 1.040 (0.017) | 1.149 (0.010) | **0.934** (**0.017**) | 29.67 (1.19) | 60.72 (0.08) | 48.91 (1.27) | **63.45** (**0.16**) | 62.32 (0.11) | 3.559 (0.070) | 2.335 (0.031) | 2.188 (0.088) | **1.203** (**0.068**) | 1.367 (0.126) |
| Retweet | 0.990 (0.016) | 1.055 (0.015) | 1.012 (0.018) | 1.663 (0.014) | **0.985** (**0.016**) | 51.72 (0.33) | **60.68** (**0.11**) | 55.35 (0.19) | 59.72 (0.11) | 60.17 (0.23) | −2.180 (0.025) | −2.597 (0.016) | −2.672 (0.023) | **-3.134** (**0.019**) | −2.897 (0.030) |
| MIMIC-II | 0.859 (0.093) | > 10 (0.114) | 1.005 (0.121) | 0.933 (0.088) | **0.810** (**0.173**) | 78.20 (5.00) | **85.98** (**2.56**) | 80.49 (5.20) | 84.30 (2.78) | 85.06 (3.65) | 1.167 (0.150) | 5.657 (0.304) | **0.939** (**0.139**) | 1.025 (0.155) | 1.354 (0.413) |
| Stack Overflow | **1.017** (**0.011**) | 1.057 (0.011) | 1.340 (0.013) | 1.052 (0.011) | 1.018 (0.011) | 53.95 (0.32) | 53.83 (0.18) | 53.00 (0.35) | **56.80** (**0.18**) | 55.58 (0.29) | 2.156 (0.022) | 2.318 (0.022) | 2.314 (0.020) | **1.873** (**0.017**) | 2.063 (0.016) |

## 6.2 COMPARISONS OF RESULTS

We compare our linear Dec-ODE with state-of-the-art TPP methods on five real-life datasets from Sec. 6.1. The summary of the overall results is in Table 1. In most cases, our approach showed comparable or better results in all metrics. It confirms that Dec-ODE successfully models the complex dynamics of MTPP with independently propagated influence functions. Specifically, Dec-ODE showed its strength in prediction tasks, represented by high accuracy and low RMSE, while ANHP was marginally better in NLL. Moreover, methods that estimate $f^*(t)$ in order to calculate $\mathbb{E}[t]$, Dec-ODE and IFL, showed a tendency to make more accurate predictions of event time.

**Discussion.** For a fair comparison, we increased the number of sampling used in the thinning algorithm (Chen, 2016; Ogata, 1981) implemented by (Yang et al., 2022). In most cases, the number of samples is multiplied by 20. However, when applied to THP on Reddit dataset the thinning algorithm was not able to correctly sample from $f^*(t)$, since it was not able to correctly sample from an intensity function with low magnitude with high fluctuation. Details are in AppendixB.4.

## 6.3 EXPLAINABILITY OF DEC-ODE

In the case of MTPP and TPP, events temporally and complexly influence $\lambda^*(t)$ and $f^*(t)$ (Hawkes, 1971; Isham & Westcott, 1979). Therefore, when considering the explainability of a neural network modeling an MTPP, it is important to understand how each event affects the prediction, and how they change through time. Dec-ODE naturally yields such information. For example, the temporal dynamics of $\lambda^*_g(t)$ and $\hat{f}^*(k|t)$ can be directly observed with the naked eyes as in Fig. 3.

Our detailed investigation on the Retweet dataset further validates the explainability of Dec-ODE. For example, some meaningful patterns from the Retweet data is visualized in Fig. 4. The Retweet data is composed of three classes: a post from a person with a small number of followers ($\sim 50\%$ of the population), a post from a person with a medium number of followers($\sim 45\%$ of the population), and a post from a person with a large number of followers ($\sim 5\%$ of the population), denoted as 0, 1, and 2, respectively. In Fig. 4(a), which visualizes $\mu(t; e_i)$ conditioned on different $e_i$, shows that each event exhibits temporally-decaying influence on $\lambda_g(t)$. Such a pattern is expected as a post on social media shows immediate reactions, rather than a time-delayed effect. Also, a post from an user with large followers exhibits slower decay. Which is natural since a post from an user with a large followers gets more exposure and results in longer lasting influence on people.

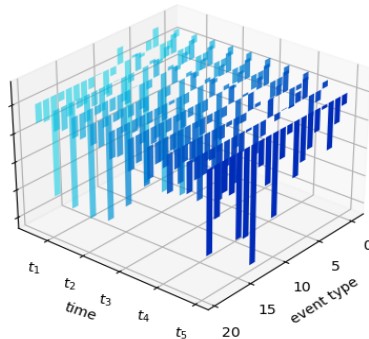

Figure 3: Visualization of propagated $\hat{f}^*(k|t, e_i)$ in StackOverflow experiment. The each axis represent time, event type, and the magnitude. The change of influence through time can be easily observed.

Moreover, the averaged proportion of influences on different mark types are illustrated in Fig. 4(b). First, even though the type 2 takes only $5\%$ of the entire population, it shows great influence in the overall occurrence of events. This may be true as a person with many followers may have a greater influence on a broader audience. Second, for each event, events with the same types have the strongest influence on each other. Considering the exponentially decaying patterns of $\mu(t)$ and the large initial magnitude of influence of type 0 shown in Fig. 4(a), we can infer that events with type

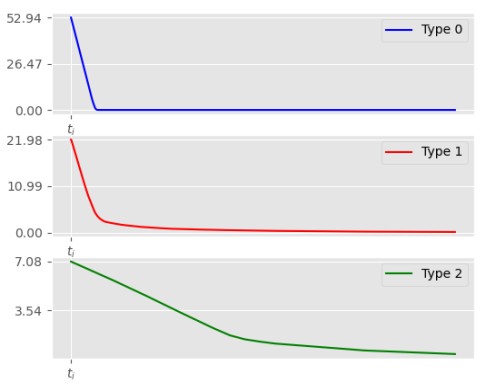


(a) Influence function $\mu(t)$ conditioned on different event types plotted on the same time range.

(b) Averaged proportion of influence from different marks on specific marks.

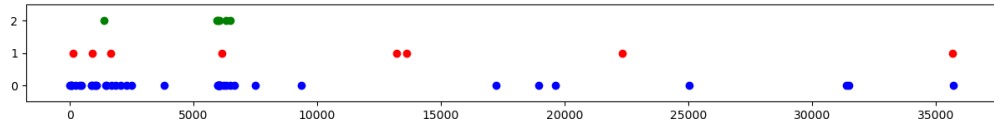

(c) Visualization of events, i.e., marks across time, affecting each other. Blue(0): users with small followers, Red(1): users with medium followers, Green(2): users with large followers.

Figure 4: Visualization of patterns found in Retweet dataset. (a) $\mu(t; e_i)$ with different marks, (b) Influence of marks on each other, (c) Actual event marks with respect to time.

0 would show a high clustering effect and actual visualization of the data in Fig. 4(c) validates our reasoning.

## 6.4 PARALLEL COMPUTING SCHEME

While Neural ODEs are effective when learning the continuous dynamics of a system, the hidden state propagation requires large computation to solve IVP step by step. In Sec. 4.3 we proposed to propagate the hidden state through time in parallel by disentangling individual events, and we validate the optimization scheme by comparing the average time for each iteration with traditional IVP. For the sequential propagation, a differential equation is solved from $t_0$ through $t_n$ step by step. Table 2 summarizes the result, where the computation time significantly decreased in all cases. Parallel computation for Dec-ODE was at least twice (for MIMIC-II) and as much as five times faster (for Reddit) than the traditional sequential computing scheme.

Table 2: Training efficiency comparison in different datasets with the average time taken for an iteration (sec/iter) as the metric. Ratio shows that the iteration time at least reduces in half.

| StackOverflow | | | MIMIC-II | | | Retweet | | | Reddit | | |
|---|---|---|---|---|---|---|---|---|---|---|---|
| parallel | Sequential | Ratio | parallel | Sequential | Ratio | parallel | Sequential | Ratio | parallel | Sequential | Ratio |
| 15.0 | 57.7 | 0.26 | 2.9 | 6.5 | 0.47 | 16.8 | 67.6 | 0.25 | 15.5 | 78.7 | 0.20 |

## 7 CONCLUSION

In this work, we presented a novel framework for modeling MTPP, i.e., Dec-ODE, that decouples influence from each event for flexibility and computational efficiency. The Dec-ODE leverages neural ODE to model continuously changing behavior of influence from individual events, which govern the overall behavior of MTPP, and we proposed a training scheme for a linear Dec-ODE, which let the influences be efficiently computed in parallel. The proposed method was validated on five real-world benchmarks yielding comparable or superior results. Moreover, it provides explainability to the modeling, which suggests significant potential for applications such as out-of-distribution detection and survival analysis.

ACKNOWLEDGMENTS

This research was supported by NRF-2022R1A2C2092336 (60%), IITP-2022-0-00290 (30%), and IITP-2019-0-01906 (AI Graduate Program at POSTECH, 10%) funded by the Korea Government (MSIT).

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

## A ADDITIONAL EXPERIMENTS

Additional experiments are performed to explore the extensibility of our work.

### A.1 LINEAR VS. NON-LINEAR INFLUENCE FUNCTION

In the table 1, the simple Dec-ODE with linear $\Phi_\lambda$ has been tested. However, $\Phi_\lambda$ and $\Phi_k$ can be any functions, even neural networks, that satisfy the conditions mentioned in 3.2 and 3.3. In fact, with linear $\Phi_\lambda$ every influence function must satisfy $\mu(t) >= 0$ to keep the intensity positive. Therefore, only an excitatory effect, where an event can only increase the intensity, can be expressed.

In this section, to navigate the extensibility of our framework, a less constrained variant is introduced and tested, denoted as *non-linear* $\Phi_\lambda$. The non-linear $\Phi_\lambda(t)$ is defined as follows:

$$\Phi_\lambda(\mu(t; e_0), \cdots, \mu(t; e_i)) = \text{softplus}\left( \sum_{e_i \in \mathcal{H}_t} \mu(t; e_i) \right). \tag{17}$$

Through this modeling approach, an inhibitory effect, where an event decreases the overall intensity, can occur.

To see how this relaxation affects in modeling TPP, both are tested in a simplified setting, where only the first 40 events of each sequence are used. The results of the experiment are summarized in Table 3 using NLL as the metric. The overall result regarding the ground intensity was improved in most situations showing that the intensity function with higher fidelity can be predicted using more flexible $\Phi_\lambda$.

However, non-linear Dec-ODE shows some limitations in predicting a time-delaying effect, where an event does not have a large influence until a certain time passes rather than having an instant influence. This happens because the inhibitory effect exceeds the excitatory effect, and the intensity function becomes 0. In such cases, the inhibitory effect rather impairs the model's ability. In conclusion, this experiment suggests that with a more flexible $\Phi_\lambda$, prediction can be made with higher fidelity in most cases. However, adding constraints has to be further discussed to robustly model different patterns in TPP.

Table 3: Experiment on non-linear $\Phi_\lambda$. The softplus is applied after the summation in order to express inhibitory effects. Details are in A.1

| StackOverflow | | MIMIC-II | | Retweet | | Reddit | |
|---|---|---|---|---|---|---|---|
| Linear | Non-linear | Linear | Non-linear | Linear | Non-linear | Linear | Non-linear |
| 0.9948 | 0.8987 | 0.2451 | 0.2183 | $-5.0872$ | $-5.1649$ | 0.5163 | 18.2571 |

### A.2 IMPUTATION

In this section, we investigate the effect of independently modeling hidden state dynamics by comparing it with methods using contextual information. Events from the StackOverflow dataset are randomly dropped to see how the behavior changes as the number of events decreases. For the fair comparison, we randomly selected 90%, 80%, 70%, and 60% indexes from the test dataset, and saved the data with the selected index as new test sets. The results using the new test sets are visualized in figure 5. The graph illustrates that the decrease in performance of Dec-ODEs shows a similar tendency when compared to others.

The result suggests that other methods cannot retrieve the loss of information. This statement can be made since Dec-ODE, which does not consider inter-event relationships, shows a similar tendency.

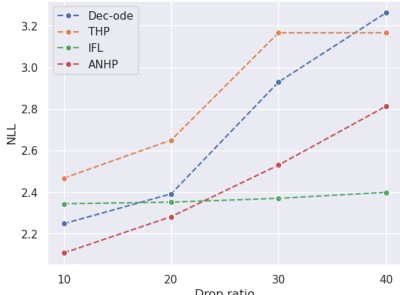

Figure 5: Imputation experiment done using Stackoverflow dataset, where from 10% to 40% of data are randomly dropped.

### A.3 TRAIN TIME COMPARISON

Table 4: Training time (sec / epoch) compared between THP, ANHP, and Dec-ODE.

|          | MOOC   | Reddit | Retweet | Stackoverflow | MIMIC-II |
|----------|--------|--------|---------|---------------|----------|
| THP      | 12.94  | 66.95  | 34.05   | 13.36         | 2.27     |
| ANHP     | 835.36 | 541.20 | 630.64  | 129.51        | 3.69     |
| Dec-ODE  | 117.35 | 242.75 | 484.08  | 123.28        | 8.12     |

Table 5: Required memory (MB) compared between baseline methods with batch size 4.

|          | MOOC        | Reddit      | Retweet     | Stackoverflow | MIMIC-II    |
|----------|-------------|-------------|-------------|---------------|-------------|
| RMTPP    | 1110 + 1114 | 1706 + 1142 | 1294 + 1112 | 1306 + 1114   | 1114 + 1106 |
| THP      | 3484        | 15304       | 1678        | 3808          | 1130        |
| ANHP     | 31310       | < 48 GB     | 11790       | 39260         | 1244        |
| Dec-ODE  | 1422        | 1472        | 1314        | 1422          | 1470        |

In order to compare the time required for training, the time required for training one epoch (sec / epoch) is measured. The experiment is conducted on a single NVIDIA RTX A6000 GPU (48GB) with the largest batch size possible for the GPU. The result can be found in the table 4. THP required the least amount of time, while methods predicting intensity at each time point showed a relatively slower training rate. In most cases, Dec-ODE shows a shorter training time per epoch.

## B IMPLEMENTATION DETAILS

### B.1 IVP SOLVING WITH VARYING TIME INTERVALS

Initial Value Problem (IVP) solving with varying time intervals is done following the method introduced in Chen et al. (2021). In short, the integration in the region $[t_{start}, t_{end}]$ is solved within $[0, 1]$ and scaled back to the original region. Using this method, every time interval with different lengths can be computed within the same number of steps. We apply this for both batch computations with different lengths and parallel computation of $h(t; e_i)$ in Sec. 4.3.

### B.2 BATCH COMPUTATION

Our method propagates $h(t)$ further than $t_N$, which is the last observed point. Therefore, in order to cover the full range of $f^*(t)$, two different masking operations are required. The first one is the one commonly used when dealing with sequences with different lengths, and we call it the *sequence mask*. With sequences with different lengths, unobserved events should be ignored for both training and inference, and the sequence mask is used to ignore unobserved time points. The second mask is the *propagation mask*, where the unobserved time points after $t_N$ are not masked. This is used for solving ODEs to propagate $h(t)$ until the decoded $\mu(t)$ converges.

### B.3 BASELINE

For THP and ANHP, we used the public GitHub repository https://github.com/yangalan123/anhp-andtt (Yang et al. (2022) with MIT License). The code for the THP has been modified by Yang et al. (2022), where time and event prediction is now made by using $\lambda(t)$ with a thinning algorithm instead of the prediction module that is parameterized by neural networks.

Moreover, as mentioned in the public GitHub repository of THP, there is an error regarding the NLL computation. In both published versions, NLL is computed conditioned on the observed history information and a current event. i.e., $L(e_i) = f(e_i|e_0, \cdots, e_i)$. Therefore, the calculation of the intensity has been corrected to $L(e_i) = f(e_i|e_0, \cdots, e_{i-1})$, where the modification is made based on the code used for the integration.

For IFL, we used the public GitHub repository https://github.com/shchur/ifl-tpp (Shchur et al. (2020), no license specified). Some modifications have been made since the original implementation does not support time point prediction. The modification was made based on the author's

comment made in the "issue" page of the repository. However, the computation of $\mathbb{E}[t]$ was not stable due to the high variance of components in the mixture model. In other words, when a distribution in a mixture model has a high variance, the exponential term in the calculation causes an overflow, which leads to an overflow of the entire calculation. Therefore, we have tightened the parameters for the gradient clipping in Shchur et al. (2020), as done in Bae et al. (2023).

### B.4 THINNING ALGORITHM

For the experiment done in Table 1, we tested different parameters for the thinning algorithm (Chen, 2016; Ogata, 1981), implemented by Yang et al. (2022), for a fair comparison. THP and ANHP sample the next time points using the thinning algorithm then take their average to compute $\mathbf{E}[t]$.

The thinning algorithm resembles the rejection sampling, in which the proposal is accepted with the probability $\lambda(t)/\lambda_{up}$ where $\lambda_{up} \geq \lambda(t)$ for all $t \in (t_{i-1}, \infty)$ (Mei & Eisner, 2017). Therefore, reliable $\lambda_{up}$ is required to sample from the correct distribution. If $\lambda_{up}$ is too high, all the samples would be rejected, whereas accept incorrect samples if $\lambda_{up}$ is too low.

In the implementation, $\lambda_{up}$ is calculated by $c * max(\lambda(s_0), \cdots, (s_m))$, where $c$ is a constant, $m$ is the number of samples used for the calculation, and $s_m$ is a uniformly sampled time point. To find correct $\lambda_{up}$, for most of the reported results in Table 1, we increased the number of $m$ by $\times 10$. When overall $\lambda(t)$ is too low, the scale goes up to 1000 in the most extreme case.

### B.5 TRAINING DETAILS

When tuning the simple Dec-ODE, three different hyperparameters are considered for the model structure with the addition of an Initial Value Problem (IVP) solving method. Three hyperparameters are the dimension of hidden state $D$, the dimension of linear layers of neural networks $N$, and the number of liner layers $L$. We haven't fully searched for the optimal parameters for each dataset. For testing, we rather applied similar parameters to all datasets.

Throughout the experiment, $D$ was chosen from $\{32, 64\}$. However, since the dynamics of $h(t)$ is highly dependent on the information in the hidden state, we expect a higher dimension would enable the neural network to learn more difficult dynamics.

The width of linear layers $N$ is chosen between $\{128, 256\}$, and the number of layers are tested between $\{3, 4, 5\}$. In most cases, the number of layers did not show much improvement in the results.

The three hyperparameters above that showed the best result are chosen. However, in most cases, $D = 64$, $N = 256$, and $L = 3$ were used.

In the case of IVP solver, two different methods were used training and testing. For training, Euler's method was used in most cases for efficiency. The purpose of training is to learn the dynamics of hidden state $h(t)$, and we believe Euler's method sufficiently satisfies such purpose, and we haven't thoroughly looked into the benefits of using different solvers.

During testing $\lambda_g(t)$, to make a more precise approximation, we utilize Runge-Kutta method with fixed step size, generally referred to as RK4. The computation of RK4 method is as follows:

$$y_{n+1} = y_n + \frac{h}{6}(k_1 + 2k_2 + 2k_3 + k_4) \tag{18}$$

$$t_{n+1} = t_n + h \tag{19}$$

where,

$$k_1 = f(t_n, y_n), \tag{20}$$

$$k_2 = f(t_n + \frac{h}{2}, y_n + h\frac{k_1}{2}), \tag{21}$$

$$k_3 = f(t_n + \frac{h}{2}, y_n + h\frac{k_2}{2}), \tag{22}$$

$$k_4 = f(t_n + h, y_n + hk_3). \tag{23}$$

The RK4 method can more precisely model the trajectory compared to Euler's method. Therefore, there is less error in the estimated $f^*(t)$. Other methods such as dopri5, RK4 with step size control, and other variants of IVP solvers can be applied for more accurate results.

To solve the IVP, we fixed the number of steps required for solving from $t_i$ to $t_{i+1}$. Similar to the IVP solver, we used a simpler setting for the training and a more precise setting for the testing. During training, in most cases, we used 16 steps in between every event. The increase in number is expected to improve the result, yet we have not thoroughly looked into it since it shows comparable results with state-of-the-art methods. During testing, we increased the number to 64.

When testing $f(k|t)$, the precise approximation of $\hat{f}(k|t, e_i)$ did not show a visible effect on the result. Therefore, for computational efficiency, Euler's method was used with the same number of step sizes used for training.

| Datasets | # of Seq. | # of Events | Max Seq. Length | # of Marks |
|---|---|---|---|---|
| MOOC | 7,047 | 389,407 | 200 | 97 |
| Reddit | 10,000 | 532,026 | 100 | 984 |
| Retweet | 24,000 | 21173,533 | 264 | 3 |
| StackOverflow | 6,633 | 480,414 | 736 | 22 |
| MIMIC-II | 650 | 2419 | 33 | 75 |

Table 6: Statistics of benchmark datasets used for comparisons.

## C  DATASET DESCRIPTION

Table 6 shows the statistics of each benchmark dataset used for testing.

### C.1  BENCHMARK DATASET

MIMIC-II and Retweet were from the public GitHub repository (Zuo et al. (2020), with no license specified). Others were also from the public GihtHub repository (Lin et al. (2022), with no license specified).

### C.2  DATA PREPROCESS

When dealing with Mooc, Reddit, Retweet, and StackOverflow, two modifications to the dataset have been made. First, when two events have the same time point, one of the events is removed. We chose the latter one in the dataset. This is because when $t_i - t_{i-1} = 0$, IFL was not able to properly estimate $f^*(t)$. Also, in many cases, TPP assumes 2 or more events happening at the same time as improbable. Second, as mentioned in Sec. 6.1, data were scaled by the standard deviation of $\{t_i - t_{i-1}\}_{i=0}^{n}$. When the scale of $t_i - t_{i-1}$ is too large, overflow happens during the calculations. Therefore, we tried to make the scale similar to the results from Bae et al. (2023), but the standard deviation was chosen since the scale was not specified in the published work.

# D VISUALIZATIONS

## D.1 VECTOR FIELD

By employing Neural ODEs (Chen et al., 2018) we can visualize the overall dynamics of functions that we estimate. Fig. 6 visualizes the vector fields. Fig. 6(a) visualizes the first dimension of the hidden state using the StackOverflow dataset. The arrow represents the dynamics where the value of the tail is given as the input. There are some visible patterns in the dynamic forms. However, a certain part of the field shows very different behavior even when they receive the same input. It shows that the hidden state dynamics learned from the data is very complex. On the other hand, $\mu(t; e_i)$ visualized in Fig. 6(b) shows a clear tendency in the region.

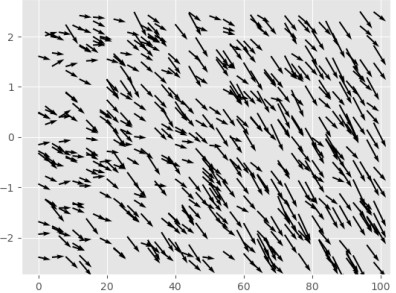
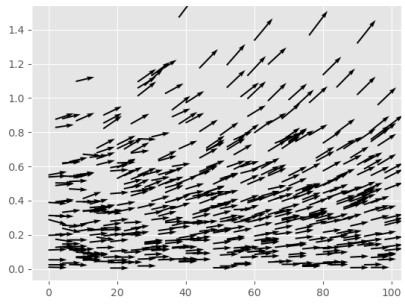

(a) Visualization of the hidden state dynamics with StackOverflow dataset.

(b) Visualization of the change of $\mu(t; e_i)$ in StackOverflow dataset.

Figure 6: Visualization of vector fields. a) is a vector field of the hidden state and b) is the vector field of $\mu(t; e_i)$. The length of the arrow represents the magnitude.

## D.2 VISUALIZATION OF CONTINUOUS DYNAMICS

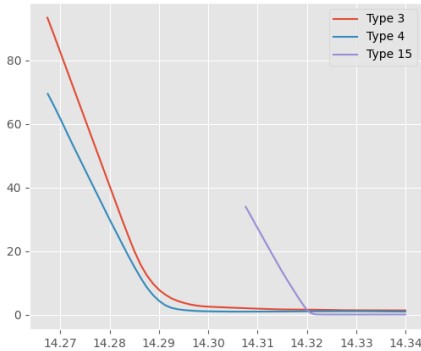

(a) Visualization of $\mu(t; e_i)$ trained using MOOC dataset.

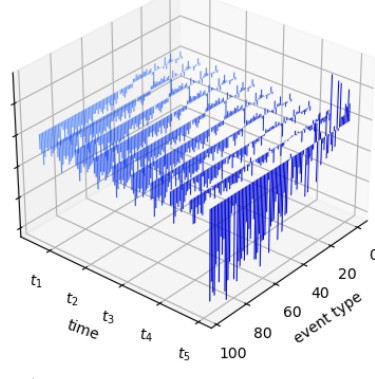

(b) $\hat{f}(t; e_i)$ changing through time, where $e_i = 14$, in MOOC dataset.

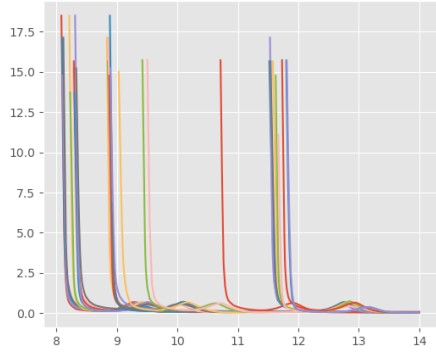

(c) $\mu(t; e_i)$ trained using Reddit dataset, each $\mu(t; e_i)$ is plotted with different colors for visibility. We can infer from the figure that there is a delayed effect in Reddit dataset.

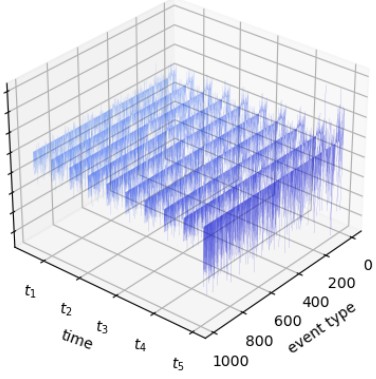

(d) $\hat{f}(t; e_i)$ trained on Reddit dataset, changing through time, where $e_i = 148$.

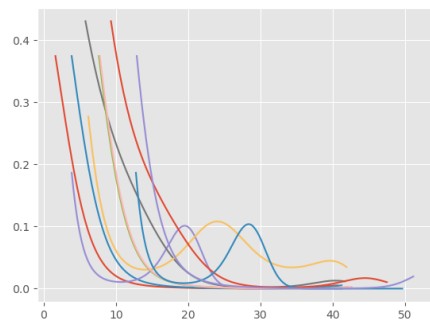

(e) Visualization of $\mu(t; e_i)$ trained using SO dataset. Each $\mu(t; e_i)$ demonstrates different dynamics.

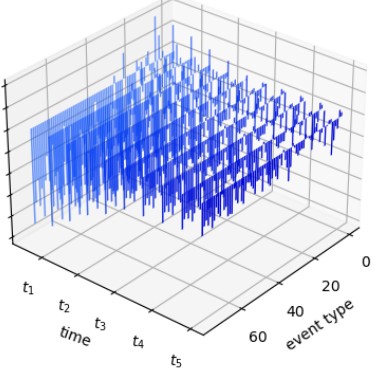

(f) Visualization of $\hat{f}(t; e_i)$ changing through time in MIMIC-II dataset, where $e_i = 10$.

Figure 7: Visualization of dynamics of $\mu$ and $\hat{f}$ in benchmark dataset.

### D.3 SIMULATION STUDY

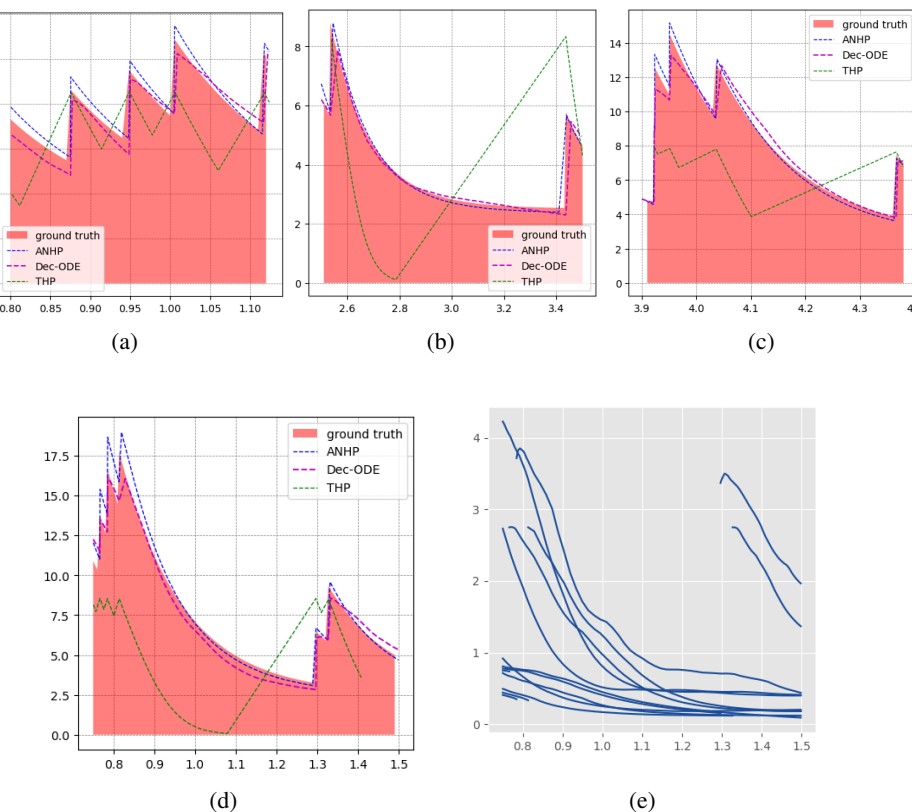

Figure 8: Visualization from a simulation study. (a), (b), (c), and (d) compare the true intensity function of a Hawkes process and the reconstructed results from THP, ANHP, and Dec-ODEs. Each $\mu(t)$ that composes the result in (d) is illustrated in (e).

|  | RMSE | ACC | NLL |
|---|---|---|---|
| THP | 0.7734 | 27.48 | 3.0121 |
| ANHP | 0.6528 | 30.31 | 0.2807 |
| Dec-ODE | **0.6568** | **30.35** | **0.2739** |

Table 7: Estimation accuracy on the simulated Hawkes process.

We have performed a simulation study on the Hawkes process following a similar procedure from the Neural Hawkes Process (NHP) (Mei & Eisner, 2017). In order to obtain the true intensity function we randomly sampled the parameters for the kernel function of the multivariate Hawkes process, and we simulated synthetic data using the tick library (Bacry et al., 2018). The range of the sampling was changed from the NHP since the scale in the library was different from the paper. In Figure 8(a) - 8(d), THP struggles to simulate the flexible dynamics of the intensity function $\lambda_g$, whereas ANHP and Dec-ODE show similar dynamics to the ground truth intensity. The table 7 also demonstrates that Dec-ODE shows better results than THP and ANHP in all metrics. The result also aligns with the results in Table 1, which suggests that Dec-ODE can simulate reliable MTPPs compared to other state-of-the-art methods. Figure 8(e) illustrates $\mu(t)$ that compose $\lambda_g(t)$ from Figure 8(d), which shows that the ground intensity $\lambda_g(t)$ can be reconstructed from the individually constructed trajectory, i.e. an MTPP can be modeled from decoupled information.

