# OpenReview forum: "Decoupled Marked Temporal Point Process using Neural Ordinary Differential Equations"
_ICLR.cc/2024/Conference — ICLR 2024 poster_

### Official Review · Reviewer_yduQ · 2023-10-30

**Soundness:** 3 good
**Presentation:** 3 good
**Contribution:** 3 good
**Rating:** 8
**Confidence:** 3

**Summary:**

This paper proposes a new Dec-ODE model to learn marked temporal point processes (MTPPs). The Dec-ODE model takes into account the individual events' influence on the underlying dynamics of the whole process and models it as a neural-ODE. Since the model is decoupled into ground intensity and conditional mark distribution, the new approach can compute the integrals for different parts in parallel, making the training faster. The experiment shows the effectiveness of the new model.

**Strengths:**

1. The proposed approach is novel. The paper takes into account the events' influence on the underlying dynamics of the later process, which is a significant factor but under-explored before.
2. The decoupled model facilitates parallel computing of the costly integrals that occur in the neural MTPP formulations.
3. The influence model gives Dec-ODE some extent of explainability as a model of MTPP.

**Weaknesses:**

1. There is a lack of comparison in computation time across different models. The parallel computing scheme is shown to reduce the computation time of neural ODE, but it is unclear whether Dec-ODE runs faster than baseline models like THP.

**Questions:**

1. I could not find the reference Yang el al., 2020 for the THP baseline. Is the correct one be [1]?

[1] Zuo, S., Jiang, H., Li, Z., Zhao, T., & Zha, H. (2020, November). Transformer hawkes process. In International conference on machine learning (pp. 11692-11702). PMLR.

---

> ### Author Response · Authors · 2023-11-16
>
> We thank the reviewer for highly appreciating the novelty and ideas of our work. We provide answers to your concerns below:
>
> - **Comparison in computation time across different models?** Please see the general response at the top where we provide evaluations for computation time per epoch.
> - **Reference for THP missing?** We appreciate the reviewer for pointing it out, and we have rectified this error with the author name and conducted a thorough review of all other references to ensure correctness.

---

### Official Review · Reviewer_Ux3F · 2023-10-30

**Soundness:** 2 fair
**Presentation:** 2 fair
**Contribution:** 2 fair
**Rating:** 5
**Confidence:** 3

**Summary:**

This paper proposes a new neural ODE-based framework for modeling marked Hawkes/mutually-exciting point processes. The framework models the intensity function (of time) as a summation of event-triggered trajectories, each of which solves a neural ODE with an initial state that depends on the time of each event and its mark. The additivity of event-triggered trajectories on the intensity function allows us for the efficient parallel computation for model training and the explainable analysis of the dependence between events, which is the advantage over previous neural network-based Hawkes processes.

**Strengths:**

- The idea of using neural ODEs to model the effect of an event on the subsequent events (essentially, triggering kernel) in temporal point processes is novel and promising.
- The parallel computing scheme of fitting in the proposed model is beneficial for practical purposes.
- The validity of the proposed model was evaluated on various real-world data.

**Weaknesses:**

- The authors insist that an advantage of the proposed method (Dec-ODE) over conventional methods (e.g., Yang et al., 2020) is that Dec-ODE can compute multiple integrations simultaneously [Section 3.2]. But Dec-ODE demands to solve ODEs to compute integrations, which is generally more time-consuming than Monte Carlo integrations needed in conventional methods. In Experiment, there are no comparative analysis about computation time (sec/iter), and the authors’ insistence about computation efficiency remains to be verified.
- The authors insist that an advantage of Dec-ODE over conventional methods (e.g., Yang et al., 2020) is the explainability due to the decoupled structure. But the decoupled structure has been adopted intensively in the literature, which were not in the benchmark models in Experiment. To verify the insistence, the authors need to compare Dec-ODE with references with the decoupled structure, which would make the pros/cons of the proposed model clearer.
- RMTPP is a standard benchmark model in the literature, and should be included in comparative experiments. Otherwise, the authors need to mention the reason for not including it.
- The explanation of the experiment setup seems to be inadequate for reproducibility. The details of neural networks (e.g., activation functions) and the machine spec used in the experiments should be shown.
- The detailed equation about how to compute the derivative of the likelihood regarding model parameter is not shown.
- Discussions about the limitation of the proposed model are not in the paper.
- To the best of my knowledge, there are sentences that seem technically inaccurate as follows, which raises a question about the paper’s quality:
	- [3rd paragraph in Section 1] the authors introduce neural Hawkes process (Omi et al., 2019) as a marked temporal point process (MTPP), but the Omi’s paper did not consider the marks of each event. Also, there are two Omi’s papers (2019a and 2019b), but they look identical.
	- [2nd paragraph  in Section 2] the authors explains that the modeling of intensity function is preferred over the pdf $f^*(t)$ due to the complex dynamics of $f^*(t)$, but the complex dynamics depending on the past events is not limited to $f^*(t)$.
	- [2nd paragraph in Section 2] $\\int_0^{\\infty} f^*(s) ds = 0$ is correctly $\\int_0^{\\infty} f^*(s) ds = 1$.
	- [3rd paragraph in Section 2] the sentence “$\\mathcal{N}_g = \\{ t_i \\}$ is a temporal point process” seems to fail to explain the definition of $\\mathcal{N}_g (t_N)$ in Eq. (2). Do the authors mean that $\mathcal{N}_g (t)$ is a counting process?
	- [Eq. (9)] the definition of $f_{\theta}(h,t)$ is not found.
	- [References] The authors of Transformer Hawkes process are not Yang et al., but Zuo et al.

**Questions:**

See Weaknesses.

---

> ### Author Response · Authors · 2023-11-16
>
> We appreciate the reviewer for the constructive comments, which will greatly help improve the paper's quality and soundness. Please see our answers to the questions below, and we hope to overturn the reviewer's decision.
>
> - **Is Dec-ODE computationally more efficient than others?** The reviewer raised a good point and we are sorry for the misleading text in Section 3.2. What we intended to argue was that computation with Neural ODEs is heavy by its nature but we mitigated the issue with the characteristics of the differential equation within our proposed framework, simultaneously solving the multi-dimensional differential equation with respect to the hidden state $h_t$. The actual computation time per epoch is given in the comment to all reviewers, where Dec-ODE takes comparable or less time than the SOTA approach. Please see the revised text in the updated manuscript highlighted in yellow starting from the bottom of page 4. We would like to make it clear that faster computation is not the advantage of our model (nor we argue it as a contribution in the introduction), but we proposed an efficient learning of a flexible model which would have taken much longer computation without our proposed optimization scheme. However, we still want to mention that numerical computation of integrals often leads to more accurate results than MCMC, which may lead to faster convergence even if sec/iter may be slower. Moreover, other methods that involve transformer do not consider the temporal nature (e.g., Markovian) of the events and need to perform integration iteratively whenever a new event is introduced.
>
> - **Decoupled structure is already well-known?** We are aware that the decoupling structure of time and event in MTPP is a general concept as we also mentioned in the Background section for MTPP. However, to the best of knowledge, decoupling individual events through separate intensity functions has been rarely studied. Through some investigation, we found a recent work VI-DPP(Panos et al., AISTATS 2023) which uses the decoupled structure of time and event. We are currently performing experiments with VI-DPP to compare decoupling behavior of Dec-ODE, whose results will be updated soon.
>
> - **Why not compare with RMTPP?** We appreciate the reviewer for pointing us to the RMTPP as a fundamental baseline. We reported the results only from the latest literature so we missed it, but we are more than happy to include RMTPP results in our paper. The results from RMTPP under our experimental setting are given in the table below which is now updated in the revised paper. The results from RMTPP compared with Dec-ODE can be found in the table below. The implementation provided by the author of RMTPP was employed, with minor adjustments made to accommodate length-varying sequences and present the results in a consistent format. The hyper parameters were found using a grid search from the list of options that can be found in RMTPP Section 6.4 paragraph 5. In most cases RMTPP showed a good performance in the time prediction task with best RMSE on StackOverflow data, and its NLL in MIMIC dataset showed a better result compared to Dec-ODE.
>
> - **Implementation details?** We feel sorry that we missed these important implementation descriptions. We utilized simple 3-layer Multi Layer Perceptrons (MLPs) for implementing the Neural ODE $\gamma()$ with linear activations. Specifically, softplus normalization on the weights is used for parameterizing $\gamma (\theta)$ as in (Chen et al., ICLR 2021). For decoders, 3-layer MLPs with ReLU activations have been used. All experiments were performed with a single NVIDIA RTX A6000.
>
> - **Details on deriving the derivative of the likelihood?** The derivative of the likelihood regarding the model parameter is calculated using the adjoint sensitivity method in (Chen et al. NeurIPS 2018), which is extensively discussed in the referenced paper. For detailed description and implementation please refer to the Neural ODEs (Chen et al. NeurIPS 2018) and the public repository (https://github.com/rtqichen/torchdiffeq).

---

> ### Author Response · Authors · 2023-11-16
>
> - **Limitations?**  A major limitation is that we assume that each $\mu(t)$ is independent from other events. Even though such an approach is beneficial for interpretability, it may be limited in learning inter-event relationships. Also, even though Dec-ODE shows comparable training time, it requires longer training time for an iteration than most conventional methods due to the flexible but complex nature of ODEs. However, notice that the actual computation time per epoch is quite reasonable and even better than ANHP (shown in the common question) given our proposed training scheme. Finally, the experimented version of the Dec-ODE can only express a self-excitatory effect, where each event only increases the intensity but not decreases, which makes the model less flexible. Actually, if we design the model to both increase and decrease the intensity simultaneously, the performance becomes better as we have shown in the Appendix A.1. However, in this case, the parallel training scheme in Section 4.3  cannot be adopted as it violates eq. (16) in the paper, and therefore we had to leave it in the Appendix.
>
> - **Technical errors?** We have corrected all the technical issues addressed by the reviewer and went through the paper again to check for other errors if there were any. These errors can be easily corrected in the text and do not affect the soundness of our model. We really appreciate the reviewer for going through the paper thoroughly to improve its quality.
>    - Reference errors for Omi et al, NeurIPS 2019 and Zuo et al, ICML 2020?: Corrected.
>    - Complex dynamics depending on the past events is not limited to $f^*(t)$?: Our objective was to underscore the challenge of modeling the probability density function $f^*(t)$ with a parametric model. As the intricate nature of the dynamics involved is quite diverse, describing these temporal variations through intensity functions allows for a more flexible modeling, often achieved with nonparametric methods.
>    - Do the authors mean that N_g(t) is a counting process?: The $N_g$ is a temporal point process with a realization $\\{ t_i \\}_{i=1} ^N$. For clarity, the text regarding the counting process in the 1st paragraph in Section 2 is removed for more precise delivery.
>    - In 2nd paragraph in Section 2, $\int f^*(t) = 0$?: It is now $\int_0 ^{\infty} f^*(t) = 1$.
>    - In Eq. (9), the definition of $f_\theta(ℎ,t)$ is not found?: It should be $\gamma (\bf{h_t}, t, \mathbf{e_i} ;\theta) $ and not $f_\theta ()$, where $\mathbf{e_i}$ is a vector of  events corresponding to $\mathbf{h_t}$.

---

> ### Author Response · Authors · 2023-11-16
>
> |  |  | MOOC |  |  | Reddit |  |  | Retweet |  |  | SO |  |  | MIMIC |  |
> |:---:|:---:|:---:|:---:|:---:|:---:|:---:|:---:|:---:|:---:|:---:|:---:|:---:|:---:|:---:|:---:|
> |  | RMSE | ACC | NLL | RMSE | ACC | NLL | RMSE | ACC | NLL | RMSE | ACC | NLL | RMSE | ACC | NLL |
> | RMTPP | 0.4732 (0.0124) | 20.98 (0.29) | -0.3151 (0.0312) | 0.9527 (0.0164) | 29.67 (1.19) | 3.5593 (0.07) | 0.9901 (0.0164) | 51.72 (0.33) | -2.1803 (0.0246) | **1.0172 (0.0112)** | 53.95 (0.32) | 2.1561 (0.0217) | 0.8591 (0.093) | 78.20 (5.0) | **1.1668 (0.15)** |
> | Dec-ODE | **0.4669 (0.0117)** | **42.08 (0.44)** | **-2.2885 (0.1913)** | **0.9335 (0.0169)** | **62.32 (1.1)** | **1.367 (0.1258)** | **0.9849 (0.0164)** | **60.22 (0.23)** | **-2.8965 (0.0304)** | 1.0372 (0.0110) | **55.58 (0.29)** | **2.0633 (0.0161)** | **0.6529 (0.1734)** | **85.06 (3.65)** | 1.3547 (0.3972) |

---

> ### Comment · Reviewer_Ux3F · 2023-11-16
> **Response to author comments**
>
> Thank you for the authors’ clarification and efforts. I stand by the point that the proposed model is promising but not clearly impactful. Still I appreciate the effort and I will increase my score to 5.
>
> **Is Dec-ODE computationally more efficient than others?** Thank you for the comparative analysis about computation time, which makes the computational pros/cons of the proposed method clearer.
>
> Note: I cannot understand why the authors mention about MCMC. The authors might to confuse Monte Carlo integration with MCMC?
>
> **Decoupled structure is already well-known?** MTPP models with decoupled structures have been actively investigated in the context of discovering latent network structures: For example, (Iwata, 2013) and (Yuan, 2019) adopted exponential decay kernels, and (Linderman, 2014) adopted logistic-normal density kernels. The idea of MTPP with decoupled structure is a straightforward extension of the classical Hawkes process, and is itself rather conventional. The comparison with VI-DPP is of course valuable, but the authors should discuss the necessity of the Neural ODE-based model against the references. Indeed, the idea of using neural ODEs to model triggering kernels in marked temporal point processes is new, but there is no value in using Neural ODE itself.
>
> **Why not compare with RMTPP?** Thank you for the analysis with RMTPP.
>
> **Technical errors?** If the authors define $\mathcal{N}_g$ as a realization of {t_i}, they should define $\mathcal{N}_g (t)$ separately, which is used in several equations. If the authors define $\mathcal{N}_g (t)$ as the total number of events occurring before $t$, then $\mathcal{N}_g (t)$ should be introduced as a counting process.

---

> > ### Author Response · Authors · 2023-11-20
> >
> > **Q1) I cannot understand why the authors mention about MCMC. The authors might to confuse Monte Carlo integration with MCMC?**
> >
> > A1) Yes, we apologize for the mistake, we were to mention the Monte Carlo integration not the MCMC.
> >
> > **Q2) MTPP models with decoupled structures have been actively investigated in the context of discovering latent network structures.**
> >
> > A2) We appreciate the noteworthy references. Unfortunately, the official implementations of the mentioned works are not available. Implementing them from scratch for the comparison would be infeasible within the discussion period. We apologize for not being able to make any update. However, please refer to the Figure 8 (e) that we have updated in the Appendix D.3. The figure illustrates the trajectory of each $\mu(t)$, where it is not in a perfectly exponential shape and has its own dynamics. We believe this is what differentiates our work from the previous decoupled structures with prior shapes, e.g., exponentioal decay, and therefore shows improvement from the baseline methods.
> >
> > **Q3) Technical errors?**
> >
> > A3) Thank you for the clarification. We have adopted the notation from Proposition 7.3.III (Daley & Vere-Jones, 2023), where $\mathcal{N}$ is defined as a marked point process, and $\mathcal{N_g}$ is defined as its ground process. However, we realize that defining $\mathcal{N_g}$ as a counting process is more clear regarding the equations in this paper. As the reviewer suggested, we have changed the definition of $\mathcal{N_g}$ as a counting process.

---

### Official Review · Reviewer_A48s · 2023-11-07

**Soundness:** 3 good
**Presentation:** 3 good
**Contribution:** 3 good
**Rating:** 6
**Confidence:** 4

**Summary:**

This paper proposes a new modeling framework for marked point processes, where the mark distribution and the conditional intensity function are treated as two separate modeling objectives. These two objectives are assumed to depend on a common latent process that is evolved through a neural ODE model. The proposed algorithm is more computationally efficient than existing methods and provides more interpretable model estimates. The effectiveness of the algorithm is demonstrated by comparisons of five benchmark data sets to state-of-art methods.

**Strengths:**

The paper is well-written and the presentation is clear. The proposed idea is sound and may have a large potential impact.

**Weaknesses:**

No simulation studies are conducted to demonstrate the estimation accuracy of the proposed algorithm when the data-generating process is assumed in the paper.

**Questions:**

The paper is quite well written, so I don't have many questions. My major concern is that no simulation studies are conducted to demonstrate the estimation accuracy of the proposed algorithm when the data-generating process is assumed in the paper. Some comparisons with existing methods are desirable as well. That way, one can have better ideas of the advantages and limitations of the proposed method, in terms of estimation accuracy, predictive accuracy, and computing times.

---

> ### Author Response · Authors · 2023-11-16
>
> We sincerely thank the reviewer for highly valuing the ideas and presentations given in our paper.
>
> - **No simulation studies?**: We are currently performing a simulation study with synthetically generated data from Neural Hwakes Process (NeurIPS 2017). We will update the results soon, which will also be discussed in the paper (or in the Appendix). Regarding the computing time, the comparisons on the real-world data are given in the general response at the top of this rebuttal.

---

> > ### Author Response · Authors · 2023-11-20
> >
> > We have performed a simulation study on the Hawkes process following the similar procedure from the Neural Hawkes Process (NeurIPS 2017). In order to obtain the truth intensity function we randomly sampled the parameters for the kernel function of the multivariate Hawkes process, and we simulated synthetic data using the tick library (https://x-datainitiative.github.io/tick/modules/hawkes.html).
> >
> > We have updated the figures that illustrate the results from the experiment in the Appendix D.3. In Figure 8 (a) - (d) THP struggles from simulating the flexible dynamics of the intensity function $\lambda_g$, whereas ANHP and Dec-ODE show similar dynamics to the ground truth intensity. The table below also demonstrates that Dec-ODE shows better results than THP and ANHP in all metrics. The result also aligns with the results in Table 1 of the main paper, which suggests that Dec-ODE can simulate reliable MTPPs compared to other state of the art methods. The Figure 8 (e) illustrates $\mu(t)$ that compose $\lambda_g(t)$ from Figure 8 (d), which shows that the ground intensity $\lambda_g(t)$ can be reconstructed from individually constructed trajectory, i.e. a MTPP can be modeled from decoupled information.
> >
> > |         | RMSE       | ACC       | NLL        |
> > |:---------:|:------------:|:-----------:|:------------:|
> > | THP     | 0.7734     | 27.48     | 3.0121     |
> > | ANHP    | 0.6540 | 29.31     | 0.2804     |
> > | Dec-ODE | **0.6398**     | **29.35** | **0.2730** |

---

### Author Response · Authors · 2023-11-16
**Comments to all Reviewers**

First of all, we would like to share our gratitude for the thorough and constructive reviews. We will try to fully address your concerns in the rebuttal below to help the reviewers agree on the acceptance of this paper. Please feel free to leave comments should you have any questions.

**Common question: Computation Time?** By the sequential solving nature of Neural-ODE, it is expected that Dec-ODE requires longer time per iteration. However, as Neural ODEs solve for the gradients using the adjoint sensitivity methods for efficient memory usage, we can handle much larger batch sizes for each iteration given the same memory resource. When training a neural network, this greatly benefits the training time in terms of epoch to go over the entire training set with robust convergence. The sec/epoch with the largest batch sizes possible for some baselines (i.e., THP and ANHP) for the GPU used in our experiment (i.e., NVIDIA RTX A6000, 48GB) is reported in the Table below, and the required memory for the same batch size (i.e., 4) across different methods are given in a separate Table. As both ANHP and Dec-ODE use neural networks to flexibly model the intensities, they require longer training time and memory usage than previous methods such as THP which interpolates intensities between observed events. Comparing Dec-ODE with the ANHP which is the state-of-the-art approach, Dec-ODE requires far less memory as well as comparable or faster computation time per epoch, where Dec-ODE was faster on the four datasets out of five.

- Computation (sec / epoch)

|         | MOOC   | Reddit | Retweet | Stackoverflow | MIMIC-II |
|:-------:|:------:|:------:|:-------:|:-------------:|:--------:|
| THP     | 12.94  | 66.95  | 34.05   | 13.36         | 2.27     |
| ANHP    | 835.36 | 541.20 | 630.64  | 129.51        | 3.69     |
| Dec-ODE | 117.35 | 242.75 | 484.08  | 123.28        | 8.12     |
|         |       |        |         |               |          |   |   |   |   |

- Required Memory for Training (MB)

|         | MOOC  | Reddit | Retweet | Stackoverflow | MIMIC-II |   |   |   |   |
|:-------:|:-----:|:------:|:-------:|:-------------:|:--------:|:---:|:---:|:---:|:---:|
| THP     | 3484  | 15304  | 1678    | 3808          | 1130     |   |   |   |   |
| ANHP    | 31310 | < 48GB | 11790   | 39260         | 1244     |   |   |   |   |
| Dec-ODE | 1422  | 1472   | 1314    | 1422          | 1470     |   |   |   |   |
|         |       |        |         |               |          |   |   |   |   |

---

> ### Author Response · Authors · 2023-11-20
>
> **Changes in Table1** There has been a slight change to the result table 1 after our careful tuning of the baseline methods, which now has been updated in the revised text. Most of the results are the same but the accuracy of ANHP in Stackoverflow data got slightly better which became the best and ours became the second best.

---

### Meta-Review · Area_Chair_Q2cu · 2023-12-06

**Metareview:**

This paper introduces an ODE-based framework for modeling marked Hawkes/mutually-exciting point processes. Two reviewers were positive about the contribution and one slightly negative. The authors made a substantial effort in addressing the concerns raised by the reviewers and, as a result, I recommend acceptance.

**Justification For Why Not Higher Score:**

One of the reviewers was mildly negative and highlighted a number of points for improvement.

**Justification For Why Not Lower Score:**

Two out of three reviewers were positive about the paper.

---

### Decision · Program_Chairs · 2024-01-16

Accept (poster)